# Looking Beyond Aggregation for Medical Federated Learning: From Analysis to Novel Architecture Design

## 1   Introduction & Related Work

Federated Learning (FL) [1] offers a privacy-preserving pathway for collaborative model development across medical institutions, making it particularly valuable for medical imaging applications where data cannot be shared [2]. However, the statistical heterogeneity of multi-center medical data, where each institution's images vary in acquisition protocols, patient populations, and disease prevalence, severely challenges FL performance [3]. Medical FL research has extensively focused on developing improved aggregation algorithms to combat this heterogeneity, meaning two other aspects of the pipeline, namely the initialization strategy and the model architecture, have remained an under-explored frontier. For initialization, we know task-relevant pre-training through self-supervised learning (SSL) is a highly-effective alternative to ImageNet (IN) pre-training [4], even more so in the data-scarce and costly annotation medical landscape, but the potential of SSL in the FL setting remains largely unexplored. In terms of architectures, it is common for FL papers to present novel aggregation methods tested on shallow/toy networks [5], which do not mirror the deep and complex architectures deployed in real-world applications. In this paper, we present a two-stage investigation to fill this gap.

First, we conducted a large-scale empirical study systematically evaluating the interplay between Architectures, Initialization strategies, and Aggregation methods (ARIAs). This study, among other findings, conclusively demonstrated that architectural choice, particularly the use of BatchNorm (BN) [6], is a dominant factor in FL performance, often outweighing the benefits of advanced aggregation algorithms. This echoes prior work which has showed BN hinders performance in heterogeneous FL due to mismatched client-specific statistics and inconsistent parameter averaging [7, 8]. In response, using other feature normalization methods like Group Normalization (GN) [9] and Layer Normalization (LN) [10] has been frequent in FL research [11, 12, 13, 14]. These alternatives slow convergence and reduce performance compared to BN [15, 16, 17].

Second, guided by these findings, we designed a BN-free architecture that combines weight standardization [18] with channel attention [19] to directly tackle the challenges posed by non-IID data. Weight standardization normalizes convolutional layer weights instead of activations, avoiding reliance on mini-batch statistics, which is problematic in FL. Channel attention generates learnable scaling factors for feature maps, suppressing features that are inconsistent across clients due to heterogeneity, and emphasizing consistent ones. By integrating channel attention with weight-standardized models, we enhance the model's ability to focus on shared, informative features across clients. Our architecture, which we name ANFR (Adaptive Normalization-free Feature Recalibration), is designed from first principles for the statistical realities of FL, providing a versatile and powerful backbone for federated medical imaging applications.

Submitted to 39th Conference on Neural Information Processing Systems (NeurIPS 2025). Do not distribute.

Table 1: Average balanced accuracy across 6 clients on Fed-ISIC. IN top-1 accuracy reported next to model name. Models listed in decreasing measured training throughput (using AMP). Difference from average balanced accuracy of centrally trained model in parentheses.

| Initialization | Random | | | ImageNet Pre-Training | | | DINO on Skin SSL dataset | | |
|---|---|---|---|---|---|---|---|---|---|
| Agg. Method | FedAvg | FedOpt | SCAFFOLD | FedAvg | FedOpt | SCAFFOLD | FedAvg | FedOpt | SCAFFOLD |
| ResNet-18 (69.76) | 51.65 (↓ 9.8) | 46.7 (↓ 14.7) | 52.45 (↓ 9) | 65.87 (↓ 4.3) | 67.55 (↓ 2.6) | 68.66 (↓ 1.5) | 66.57 (↓ 5.7) | 62.36 (↓ 10) | 66.87 (↓ 5.4) |
| NF-ResNet-50 (80.64) | 55.93 (↓ 6.1) | 56.25 (↓ 5.8) | **59.64 (↓ 2.4)** | 71.88 (↑ 0.9) | 68.75 (↓ 2.2) | 71.53 (↑ 0.5) | 67.83 (↓ 0.7) | 67.92 (↓ 0.6) | 70.11 (↑ 1.6) |
| ResNet-50 (80.86) | 49.11 (↓ 12) | 46.91 (↓ 14.2) | 48.13 (↓ 13) | 67.97 (↓ 6.3) | 66.16 (↓ 8.1) | 68.48 (↓ 5.8) | 65.16 (↓ 7.2) | 66.46 (↓ 5.9) | 66.34 (↓ 6) |
| WRN-50-2 (81.6) | 50.53 (↓ 8) | 50.12 (↓ 8.4) | 51.03 (↓ 7.5) | 69.54 (↓ 5.3) | 67.68 (↓ 7.2) | 70.34 (↓ 4.5) | 65.56 (↓ 6.9) | 64.22 (↓ 8.3) | 66.66 (↓ 5.8) |
| DenseNet-121 (74.43) | 49.42 (↓ 13.3) | 45.95 (↓ 16.8) | 52.79 (↓ 9.9) | 67.34 (↓ 5.8) | 68.03 (↓ 5) | 68.52 (↓ 4.6) | 66.28 (↓ 5.3) | 64.94 (↓ 6.6) | 67.38 (↓ 4.2) |
| SWIN-T (81.47) | 45.73 (↑ 23.2) | 44.13 (↑ 21.6) | 45.00 (↑ 22.5) | 71.19 (↓ 1.3) | 71.81 (↓ 0.6) | 73.13 (↓ 0.7) | 72.13 (↑ 1.7) | 71.40 (↑ 0.9) | 73.77 (↑ 3.3) |
| EfficientNetV2-S (84.22) | 46.59 (↓ 10.8) | 46.59 (↓ 10.8) | 47.51 (↓ 9.8) | 70.00 (↓ 9.6) | 71.48 (↓ 8.1) | 73.18 (↓ 6.4) | 57.99 (↓ 14.9) | 59.74 (↓ 13.1) | 64.98 (↓ 7.9) |
| ViT-B-16 (81.07) | 47.84 (↑ 7.2) | 49.52 (↑ 8.9) | 48.44 (↑ 7.8) | 65.86 (↑ 1.6) | 65.18 (↑ 0.9) | 68.09 (↓ 3.8) | 71.06 (↓ 2.9) | 71.52 (↓ 2.5) | 69.49 (↓ 4.5) |
| ConvNext-S (83.61) | 48.10 (↓ 7.9) | 49.93 (↓ 6.1) | 48.56 (↓ 7.5) | **75.08 (↓ 0.1)** | 73.40 (↓ 1.7) | 74.28 (↓ 0.8) | 72.07 (↓ 3) | 73.57 (↓ 1.5) | **74.56 (↓ 0.5)** |

Table 2: Average accuracy across 4 clients on OrganAMNIST with $\alpha = 0.1$. IN top-1 accuracy reported next to model name. Models listed in decreasing measured training throughput (using AMP). Difference from the accuracy of the centrally trained model in parentheses.

| Initialization | Random | | | ImageNet Pre-Training | | | DINO on Abdomen-SSL | | |
|---|---|---|---|---|---|---|---|---|---|
| Agg. Method | FedAvg | FedOpt | SCAFFOLD | FedAvg | FedOpt | SCAFFOLD | FedAvg | FedOpt | SCAFFOLD |
| ResNet-18 (69.76) | 88.8 (↓5.6) | 90.76 (↓3.6) | 89.16 (↓5.2) | 94.02 (↓1.9) | 94.78 (↓1.2) | 94.33 (↓1.6) | 83.54 (↓9.8) | 87.89 (↓5.5) | 84.76 (↓8.6) |
| NF-ResNet-50 (80.64) | 71.6 (↓16.3) | 78.84 (↓9.1) | 73.8 (↓14.1) | 94.39 (↓1.4) | 95.26 (↓0.5) | 95.2 (↓0.6) | 84.58 (↓7.9) | 87.93 (↓4.5) | 86.92 (↓5.5) |
| ResNet-50 (80.86) | 83.32 (↓10.5) | 86.6 (↓7.2) | 84.82 (↓9.0) | 91.98 (↓3.5) | 92.98 (↓2.5) | 92.32 (↓3.1) | 81.33 (↓12.9) | 85.69 (↓8.5) | 81.49 (↓12.8) |
| WRN-50-2 (81.6) | 84.52 (↓9.6) | 85.58 (↓8.5) | 83.82 (↓10.3) | 90.56 (↓4.3) | 91.71 (↓3.2) | 90.4 (↓4.5) | 79.98 (↓13.7) | 85.02 (↓8.6) | 77.09 (↓16.5) |
| DenseNet-121 (74.43) | 86.01 (↓8.6) | 89.12 (↓5.5) | 85.06 (↓9.6) | 94.72 (↓2.2) | 95.1 (↓1.9) | 94.68 (↓2.3) | 85.26 (↓9.2) | 89.21 (↓5.3) | 84.94 (↓9.5) |
| SWIN-T (81.474) | 83.03 (↓8.6) | 85.17 (↓6.4) | 83.16 (↓8.4) | 95.64 (↓0.6) | 95.83 (↓0.4) | 95.83 (↓0.4) | 83.4 (↓8.2) | 86.4 (↓5.2) | 84.8 (↓6.8) |
| EfficientNetV2-S (84.22) | 88.8 (↓6.2) | **91.46 (↓3.6)** | 89.19 (↓5.9) | 94.0 (↓2.7) | 94.26 (↓2.4) | 93.46 (↓3.2) | 61.19 (↓31.6) | 67.54 (↓25.3) | 56.2 (↓36.6) |
| ViT-B-16 (81.072) | 83.14 (↓4.2) | 83.52 (↓3.9) | 83.85 (↓3.5) | 95.3 (↓1.5) | 95.96 (↓0.9) | **96.01 (↓0.8)** | 81.34 (↓6.8) | 83.76 (↓4.4) | 81.99 (↓6.2) |
| ConvNext-S (83.61) | 53.76 (↓35.4) | 56.07 (↓33.1) | 55.34 (↓33.8) | 94.12 (↓2.6) | 94.92 (↓1.8) | 94.84 (↓1.9) | 87.31 (↓6.0) | **89.68 (↓3.7)** | 87.64 (↓5.7) |

## 2 Methodology

To quantitatively understand architectural impacts in FL, we conducted an exhaustive benchmark evaluating 9 modern architectures, spanning convolutional networks (ResNet-18/50 [20], Wide-ResNet-50-2 [21], DenseNet-121[22], NF-ResNet-50[23], EfficientNetV2-S[24], ConvNext-S[25]) and transformers (ViT-B-16[26], SWIN-T[27]) across three initialization strategies (random weights, IN pre-training, domain-specific SSL) and three aggregation methods (FedAvg[1], FedOpt[12], SCAFFOLD[28]). We evaluated these ARIAs on the tasks of skin lesion classification on Fed-ISIC2019[29] and abdominal organ classification on OrganAMNIST[30]. For the SSL component, our *Skin-SSL* pretraining dataset was created from 3 skin lesion datasets [31, 32, 33], while the *Abdomen-SSL* dataset was created by extracting 20 slices around the center of each volume in 4 abdominal CT datasets[34, 35, 36], cropping around the subject, resizing to 224x224 and copying the channel over, resulting in 21,000 whole abdomen images.

Next, guided by the ARIA findings, we developed ANFR as an architectural solution specifically designed for FL's statistical challenges. ANFR eliminates dependency on batch-specific statistics through two synergistic components, Scaled Weight Standardization and Adaptive Feature Recalibration. 1) Scaled Weight Standardization (SWS)[23]: instead of normalizing activations, ANFR normalizes convolutional weights themselves using carefully scaled standardization that maintains signal propagation stability. This ensures consistent forward passes regardless of client-specific data distributions, removing the statistical conflicts inherent in BN during federation. 2) Adaptive Feature Recalibration: to actively combat heterogeneity and compensate for lost regularization, we integrate channel attention mechanisms after weight-standardized layers. This enables dynamic suppression of client-specific noisy features while amplifying universally informative patterns. We validate ANFR on Fed-ISIC2019, FedChest (our own multi-label chest X-ray dataset with 4 clients and covariate shift), and CIFAR-10 [37]. To benchmark ANFR, we perform focused, ablated studies against strong baselines: BN-ResNet, GN-ResNet, SE-ResNet [19], and NF-ResNet[23]. All models are evaluated under multiple aggregation methods (FedAvg, FedProx, SCAFFOLD, FedAdam).

## 3 Results

**Finding 1: Architecture Dominates Performance.** Architecture choice yielded up to 30% performance differences while aggregation methods typically changed results by <2%. On Fed-ISIC with ImageNet initialization, ConvNeXt-S achieved 75.08% while ResNet-50 reached only 67.97%. The

Table 3: Performance comparison across all architectures under different global FL aggregation methods and different datasets. Best in bold, second best underlined. ANFR consistently outperforms the baselines, often by a wide margin.

| Dataset | Method | Architecture | | | | |
|---|---|---|---|---|---|---|
| | | BN-ResNet | GN-ResNet | SE-ResNet | NF-ResNet | ANFR (Ours) |
| Fed-ISIC2019 | FedAvg | 66.01±0.73 | 65.09±0.42 | 65.29±1.32 | 72.49±0.60 | **74.78±0.16** |
| | FedProx | 66.49±0.41 | 66.51±1.21 | 66.29±0.63 | 71.28±2.14 | **75.61±0.71** |
| | FedAdam | 65.88±0.67 | 64.60±0.39 | 65.18±1.90 | 69.96±0.14 | **73.02±0.93** |
| | SCAFFOLD | 65.41±0.72 | 68.84±0.46 | 68.99±0.18 | 73.30±0.50 | **76.52±0.60** |
| FedChest | FedAvg | 82.80±0.13 | 83.40±0.25 | 82.14±0.18 | 83.40±0.11 | **83.49±0.14** |
| | FedProx | **82.14±0.10** | 82.04±0.08 | 81.50±0.26 | 81.26±0.58 | **82.14±0.10** |
| | FedAdam | 83.02±0.11 | 82.11±0.10 | 82.72±0.16 | 83.10±0.09 | **83.33±0.07** |
| | SCAFFOLD | 83.52±0.14 | 83.95±0.05 | 83.50±0.08 | 84.06±0.02 | **84.26±0.10** |
| CIFAR-10 | FedAvg | 91.71±0.74 | 96.60±0.11 | 94.07±0.04 | 96.72±0.05 | **97.42±0.01** |
| | FedProx | 95.03±0.04 | 96.05±0.04 | 94.60±0.07 | **96.82±0.04** | 96.33±0.09 |
| | FedAdam | 91.23±0.29 | 95.80±0.24 | 94.09±0.17 | 95.54±0.10 | **96.93±0.06** |
| | SCAFFOLD | 92.51±0.99 | 96.78±0.01 | 94.30±0.03 | 96.84±0.01 | **97.38±0.03** |

implication is clear: selecting the right architecture provides much larger gains than sophisticated aggregation algorithms.

**Finding 2: ImageNet Initialization Generally Wins, But SSL Shows Promise.** ImageNet consistently provided best results but medical SSL demonstrated domain-specific value: Skin-SSL achieved 74.56% on Fed-ISIC, nearly matching ImageNet's 75.08%. Critically, SSL enables FL for non-standard medical images (non-RGB, varying resolutions) without introducing aliasing artifacts from forced resizing. For transformers, pre-training proved essential: random initialization yielded 45-48% on Fed-ISIC versus 65-73% with proper initialization.

**Finding 3: Normalization Layers Create FL Bottlenecks.** Batch Normalization's limitations were stark. Normalization-Free ResNet-50 consistently outperformed standard ResNet-50: on Fed-ISIC with random initialization, 55.93% versus 49.11% . Under heterogeneity, BN models suffered 15-20% performance drops while NF models remained stable. This robustness stems from avoiding client-specific statistics that become meaningless when averaged across heterogeneous distributions.

**Finding 4: Model Scaling Paradox.** Contrary to centralized learning intuitions, deeper/wider models underperformed in FL. ResNet-18 beat ResNet-50 in 7/9 Fed-ISIC experiments despite fewer parameters. Wide-ResNet-50-2 (2.7× parameters) showed minimal gains over ResNet-50. DenseNet-121 achieved competitive performance with 68% fewer parameters than ResNet-50. This suggests FL favors architectural efficiency over raw capacity—likely because larger models are harder to optimize under non-IID conditions.

**Finding 5: Aggregation Methods Have Limited but Consistent Effects.** SCAFFOLD provided modest improvements (∼1.3% on Fed-ISIC) but required 3× communication overhead. FedOpt showed mixed results: helping heterogeneous OrganAMNIST (+2.4%) but hurting Fed-ISIC (-0.59%), with high sensitivity to server learning rate. Remarkably, simple FedAvg remained highly competitive, often achieving the best results. The marginal gains from complex aggregation methods further emphasize that architectural improvements offer more promising returns.

**Finding 6: ANFR provides universal performance benefits.** As see on 3, our proposed architecture manages to consistently outperform the baselines across all datasets and aggregation methods. This serves as clear evidence more research into architectures for FL can advance the field just as much, if not more than, aggregation methods.

The results yielded a clear verdict: architectural selection consistently outweighed aggregation improvements, with networks employing Batch Normalization suffering dramatic performance drops up to 14% in balanced accuracy under heterogeneous conditions. This systematic analysis identified BN's dependency on consistent batch statistics as a fundamental architectural weakness in the FL setting, motivating our architectural redesign.

## 4 Potential Negative Societal Impact

While our study provides valuable insights, there are potential negative impacts to consider. The emphasis on performance metrics could overshadow critical considerations such as model fairness across different patient demographics, interpretability for clinical decision-making, and robustness to distribution shifts in real-world medical settings. Finally, while SSL pre-training shows promise for non-ImageNet domains, the requirement to create large-scale medical SSL datasets may be infeasible for rare diseases or under-represented populations, potentially widening healthcare disparities .

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
