# OpenReview forum: "Looking Beyond Aggregation for Medical Federated Learning: From Analysis to Novel Architecture Design"
_EurIPS.cc/2025/Workshop/MedEurIPS — EurIPS 2025 Workshop MedEurIPS Submission_

### Official Review · Reviewer_ovzt · 2025-10-31
**Architecture vs. Aggregation in FL**

**Rating:** 6
**Confidence:** 4

**Review:**

Large empirical study on federated learning (architectures, initialization, aggregation) and a BN-free backbone (ANFR) combining weight standardization + channel attention.

Strengths:
- Clear empirical setup and relevant takeaway
- ANFR is simple, reproducible, and improves over baselines.

Weaknesses:
- FedOpt and SCAFFOLD are standard baselines but not state-of-the-art; more recent approaches (e.g., FedDG, MOON, FedDyn) could provide a stronger context.
- The claims could also be more carefully scoped, as the relative influence of architecture versus aggregation likely depends on the severity and type of non-IID data;

---

### Official Review · Reviewer_3YHM · 2025-11-03
**Review comments**

**Rating:** 6
**Confidence:** 4

**Review:**

This paper presents a systematic large-scale study on the often-overlooked interplay between Architectures, Initialization, and Aggregation in medical Federated Learning. The finding that architectural choice (particularly Batch Normalization) is a dominant factor, often outweighing complex aggregation methods, offers important practical guidance for the field.

Minor concerns:
1. The author should include a proper abstract section.
2. Channel attention is common operation that has been widely studied over pastyears, and the motivation for incorprating this is not clear and not well demonstrated.

---

### Decision · Program_Chairs · 2025-11-03

**Decision:**

Accept (Poster)

**Comment:**

Both reviewers agree that the paper presents a valuable large-scale empirical study on the interplay between architecture, initialization, and aggregation in medical federated learning.